**Data Availability Statement:** Raw data used in this study belong to European Union's Research Executive Agency (REA) and are not publicly

# Ethics appraisal procedure in 79,670 Marie Skłodowska-Curie proposals from the entire European HORIZON 2020 research and innovation program (2014–2020): A retrospective analysis

**Ilse De Waele**[1], **David Wizel**[1], **Livia Puljak**[2], **Zvonimir Koporc**[2]*

**1** European Union's Research Executive Agency, Brussels, Belgium, **2** Center for Evidence-Based Medicine and Healthcare, Catholic University of Croatia, Zagreb, Croatia

* zvonimir.koporc@unicath.hr

## Abstract

### Introduction

Horizon 2020 was the most significant EU Research and Innovation programme ever implemented and included the Marie Skłodowska-Curie Actions (MSCA). Proposals submitted to the MSCA actions awere subject to the Ethics Appraisal Procedure. In this work we explored the ethics appraisal procedure in MSCA H2020.

### Methods

Using a retrospective analysis of pooled anonymized data, we explored the ethics appraisal procedure on proposals submitted to Marie Skłodowska-Curie Actions (MSCA) during the entire Horizon 2020 program period (N = 79,670).

### Results

Our results showed that one of the most frequently identified ethics categories was Data protection. We also detected slight differences between applicants' and the ethics reviewers' awareness of ethical issues. Trajectory analysis of all ethics screened proposals appearing on main lists showed that a minimal portion of all screened submissions required additional ethics checks in the project implementation phase.

### Conclusion

Personal data protection is one of the most represented ethics categories indicated among MSCA actions which exhaust ethics assessment efforts and may lead to "overkills" in ethics requirements. Excluding the majority of personal data protection assessment from the ethics assessment, except for parts which are directly related to ethics like "Informed consent procedures", might be necessary in the future. A gap in understanding of ethics issues

available. This manuscript used data on ethics requirements for submitted proposals under Horizon 2020 MSCA calls (2014-2021). The proposals were submitted to the Research Executive Agency, established by the European Commission (REA). The first and the second author of the publication are employed by REA and, therefore, since the REA is the primer data owner, the source of the data is not to be considered as a third party. Permission from the first and the second author was granted to the third and fourth author (LP and ZK) to design the proposed study and to use the summarized data prepared by the first and the second author. The first and the second author are employees of the REA and had access to raw data, since it is the property of their employer (REA). Therefore, access to the data is restricted. The third and the fourth author (LP and ZK) had access only to a pooled anonymized data and did not receive any special privileges that other researchers would not have. Contact for interested researchers to request data access can be made through the Research Enquiry Service of the European Commission (URL: https://ec.europa.eu/info/research-and-innovation/contact/research-enquiry-service_en).

**Funding:** This study was funded by the project Promoting integrity in the use of research results in evidence-based policy: a focus on non – medical research (PRO-RES), Grant agreement ID: 788352, funded by the EU funding line SwafS-21-2017 - Promoting integrity in the use of research results in evidence based policy: a focus on non-medical research.

**Competing interests:** The authors have declared that no competing interests exist.

between applicants and reviewers' points to the necessity to further educate researchers on research ethics issues.

## Introduction

Ethics and integrity in research practices are essential to ensuring professional standards are maintained and that the subjects and users of research can be assured of maximized benefits and minimal harms [1]. Whether research is conducted in health and life sciences [2] or social sciences and humanities [3], the output of scientific research should never lead to distrust in the integrity of research results and the authors of such results [4]. However, it remains the case that unethical, questionable research practices and scientific misconduct are still present in the academic community worldwide [5]. Emerging technologies such as CRISPR technology [6] give rise to unanticipated and previously unrecognized potential ethical issues. National legislation may not promptly keep pace with such a fast and rapid development of new technologies, and, also, there is a different approach in the legislation of such developments in different countries, even at the EU level. An example of such legal heterogeneity is human embryonic stem cells (hESC) use for research purposes [7].

It is for such reasons that the European Commission (EC) makes tremendous efforts to secure the most efficient Ethics Appraisal/Review Procedure and provides extensive support and guidance to help to researchers during the evaluation of EC-funded research proposals. Even though all European research must follow the European Code of Conduct for Research Integrity [8], the EC additionally publishes numerous ethics guidance materials [9–15] and maintains a dedicated website for ethics in projects funded under the HORIZON 2020 framework [16].

Horizon 2020 was the most significant EU Research and Innovation programme ever implemented, which had nearly €80 billion of funding available over the period of 7 years (2014 to 2020). Horizon 2020 will be succeeded by the new framework programme HORIZON EUROPE, which will be running from 2021–2027. Horizon 2020 had multiple sections, including the Marie Skłodowska-Curie Actions (MSCA), which provided grants for researchers of all career stages and encouraged transnational, intersectoral, and interdisciplinary mobility. With these grants, researchers had an opportunity to move between academic and other settings in all fields of research and innovation [17].

There are five types of MSCA actions: Innovative Training Networks (ITN), which enable the formation of research networks and joint training and doctoral programs, Individual Fellowships (IF) that will allow the mobility of experienced researchers between countries, Research and Innovation Staff Exchanges (RISE) that support the short-term mobility of research and innovation personnel at all career levels, COFUND programs that enable co-funding of regional, national and international programs to finance fellowships that include mobility to or from another country, and The European Researchers' Night (NIGHT), a program that funds organization of a public science-promotion event called Researchers' Night [18]. COFUND, NIGHT, RISE and ITN are institutional applications while only IF applications are submitted from the side of individual researchers with the support of their institutions.

Proposals submitted to the MSCA actions are evaluated using peer-review scientific experts [19]. Additionally, all proposals submitted to Horizon 2020 were subject to the Ethics Appraisal Procedure, which starts with a preliminary Ethics Screening, usually by one expert. If needed as a consequence of possible sensitivities or ethical concerns in the next step, an Ethics Assessment with three or more experts is conducted. The Ethics Review Procedure can lead to additional

ethics requirements that become contractual obligations and must be respected and fulfilled from the applicant's side [16]. An Ethics Check may be required for proposals with complex or sensitive issues which would be conducted once the project was up and running. The prevalence of specific ethics issues self-reported by the applicants and identified by experts in research proposals funded by EC, and their changes over time, is largely unknown.

This study aimed to analyse ethics issues in the MSCA proposals submitted from 2014 to 2020, and to explore any differences between applicants' awareness and the opinions of expert ethics peer-reviewers conducting the review. We conclude with a discussion and interpretation of the implications of that analysis.

## Materials and methods

### Study design

This was a retrospective analysis.

### Ethics statement

The study protocol was not subject to the institutional review board assessment because the study used anonymized portions of protocols submitted to REA. Since the proposal submission and proposal assessment workflow is entirely electronic, the REA's electronic system allows extracting partial information from the proposals. Thus, only information about the type of proposal and details of ethics assessment were analyzed by the first author for this study (employee of REA). The REA has endorsed this approach, which waives the requirement for informed consent, and approved the study protocol.

### Analysed data

For the purpose of this study, data were received by courtesy of the Research Executive Agency of the European Commission (EC) from the EC statistic tool called 'Corda' (for the whole H2020 period 2014–2020). Regarding the terminology, the difference is maintained between terms 'proposal' and 'project'. Only submissions chosen to sign the grant agreement at the end of the evaluation process were considered as projects; all others are considered proposals [6]. Some ethics categories are shortened and appear differently in Corda than in the H2020 Ethics self-assessment table (e.g., ENVIRONMENT PROTECTION QUESTION vs. Environment, health & safety). Terms used in Corda are thus used in figures and in tables of this manuscript.

The data analysed in this study included proposals evaluated under MSCA, including actions COFUND, IF, ITN, RISE, evaluated within the Horizon 2020 (H2020) EU Framework programme for research, during 2014–2020. Since the NIGHT is a Coordination and Support Action (CSA), for the NIGHT, we only showed the number of proposals submitted under this call and excluded it from the further analysis.

Withdrawals and duplicates were removed from the counts. Withdrawals were proposals officially submitted to a given call by the call deadline, and on the initiative/request of the applicant, removed from an ongoing evaluation process. Duplicates are identical proposals submitted in error and not to be evaluated.

Inadmissible proposals refer to those that failed to adhere to the required administrative standards and cannot be evaluated—e.g., a certain document is missing. Ineligible proposals do not meet all eligibility criteria as defined in the WP for that call and cannot be evaluated—e.g., a researcher is not experienced.

We analysed the total number of proposals submitted and the number of proposals retained on the main list.

The main list is defined as the proposals that passed evaluation with a sufficiently high score to be funded with the immediately available budget for the call (and specific ranking list if applicable).

The reserve list is defined as proposals that passed evaluation with a score not high to be funded with the immediately available budget for the call (and specific ranking list if applicable). They can, however, be activated should an applicant on the main list drop out and funds become available.

The frequency of proposals for which the applicants declared any issues in the ethics self-assessment table was assessed in all proposals and in the main list by the ethics experts after the ethics screening.

The frequency of eleven categories of issues identified on the Ethics Issues Table (EIT) and used in the MSCA ethics evaluations was assessed in all proposals and in the main list. Trends were presented as figures; source data for all frequencies and percentages were presented in the supplementary files. These categories include (in alphabetical order): 'Animals'; 'Dual use'; 'Environment/health & safety'; 'Exclusive focus on civil applications'; 'Human cells / tissues'; 'Human embryos & foetuses'; 'Human beings'; 'Potential misuse of research results'; 'Non-EU countries'; 'Protection of personal data'; and 'Other ethics issues'. An additional category 'General' was used for ethics requirements that do not fall into any other category. It also includes information on an ethics check should that be required, certain general approvals, use of ethics mentors, etc.

For the comparison of ethics issues declared in applicants' ethics self-assessment forms and issues identified by ethics experts, only percentages of the most commonly used ethics categories (those used in more than 5% of the proposals) were presented in figures of the main text, while all frequencies and percentages were presented in supplementary files. For this analysis, values were presented in the main manuscript for IF, ITN and RISE action. Data for the COFUND action were presented only in the supplementary data since at the year 2015 ethics evaluation process for the COFUND action was changed only to the YES/NO determination of ethics issues, which consequently led to values of 0% for five ethics categories ('Dual use'; 'Human embryos & foetuses'; 'Other ethics issues'; 'Exclusive focus on civil applications' and 'Potential misuse') in the time frame of 2015–2020 which are then excluded from figures. Similar loss of values (0%) in the same period applies for the COFUND ethics checks.

For all proposals, we analyzed the 'ethics trajectory', the final summary of the process, indicating what the applicant declared, what happened during the screening/review, and whether or not there was an ethics check proposed.

The authors of the manuscript did not access the entire proposals submitted to the REA. Since the proposal submission and proposal assessment workflow is entirely electronic, the REA's electronic system allows extracting partial information from the proposals. Thus, only information about the type of proposal and details of ethics assessment was extracted and analyzed for this study. Thus, the authors were unable to match those data with any identifying information of the Applicants. The REA has endorsed this approach and approved the study protocol.

Data were presented according to the submission year, action and panel.

## Results

### Number of submitted proposals and their success rate

In the analysed period, 79670 proposals were submitted to the MSCA, with the majority of proposals belonging to IF and ITN (Fig 1). Among the submitted proposals, 11274 (14.3%) were chosen for the main list across the entire framework programme.

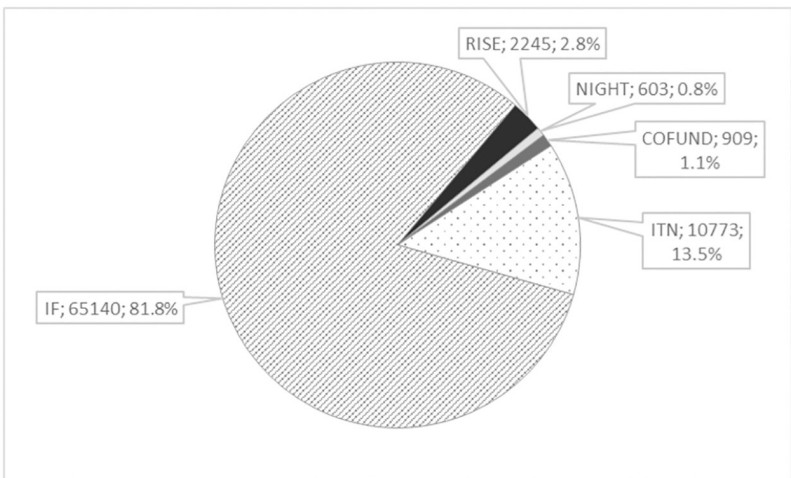

**Fig 1. Number of proposals submitted to different Marie Sklodowska Curie actions from 2014 to 2020.**

## Ethics self-assessment by the applicants in all proposals

From the 79670 submitted proposals, we did not further analyze those for the NIGHT action (N = 603). Among the remaining 79067 proposals analyzed in detail, in 34339 (43.4%) proposals, the applicants declared one or more ethics issues. This percentage of issues reported in the ethics self-assessment table by the applicant ranged from 40.5% in the year 2016 to 47.0% in year 2014 (Fig 2) in the analyzed period.

## Categories of issues reported in ethics self-assessment table by years and actions

The frequency of eleven categories of issues reported by the applicants in ethics self-assessment in all the submitted proposals over the years is shown in Table 1. The data indicate that the most commonly used categories by applicants in self-assessment were related to the use of humans, human cells/tissues and animals in research, the inclusion of non-EU countries, and protection of personal data (Table 1).

Ethics issues related to non-EU countries substantially decreased after 2015. The frequency of 'Environment, health and safety' issues remained stable until 2018, when a slight decrease is

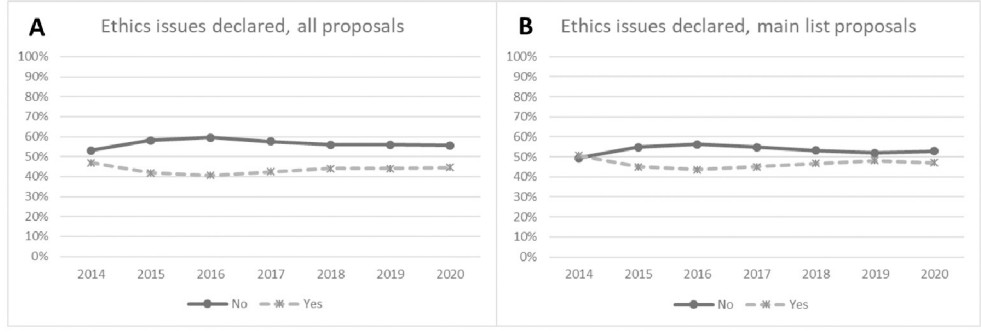

**Fig 2. Percentage of A) proposals submitted to Marie Sklodowska Curie actions from 2014 to 2020, and B) proposals on the main list with issues reported in ethics self-assessment table by the applicant.**

**Table 1. Frequency of ethical issues reported by the applicants in self-assessment in all the submitted proposals submitted to Marie Skłodowska-Curie actions during the period 2014–2020.**

| Ethics issues declared in proposal, by category | 2014 | | 2015 | | 2016 | | 2017 | | 2018 | | 2019 | | 2020 | |
|---|---|---|---|---|---|---|---|---|---|---|---|---|---|---|
| | % | # | % | # | % | # | % | # | % | # | % | # | % | # |
| PROTECTION OF PERSONAL DATA | 15,28% | 1162 | 16,70% | 1368 | 19,05% | 1562 | 19,72% | 1748 | 20,36% | 2003 | 21,23% | 2116 | 21,88% | 2537 |
| NON-EU COUNTRIES | 25,49% | 1938 | 17,64% | 1445 | 12,44% | 1020 | 11,85% | 1050 | 10,82% | 1065 | 11,21% | 1117 | 12,48% | 1447 |
| ENVIRONMENT PROTECTION QUESTION | 7,46% | 567 | 8,36% | 685 | 7,22% | 592 | 8,21% | 728 | 10,07% | 991 | 9,60% | 957 | 10,11% | 1172 |
| OTHER ETHICS ISSUES | 0,99% | 75 | 0,83% | 68 | 1,30% | 107 | 1,07% | 95 | 0,97% | 95 | 1,05% | 105 | 1,01% | 117 |
| DUAL USE | 0,32% | 24 | 0,32% | 26 | 0,26% | 21 | 0,17% | 15 | 0,19% | 19 | 0,15% | 15 | 0,17% | 20 |
| HUMANS | 18,78% | 1428 | 20,95% | 1716 | 23,44% | 1922 | 22,77% | 2018 | 23,04% | 2267 | 24,00% | 2392 | 23,73% | 2751 |
| HUMAN CELLS / TISSUES | 11,53% | 877 | 13,38% | 1096 | 14,12% | 1158 | 14,23% | 1261 | 14,29% | 1406 | 13,89% | 1384 | 12,84% | 1489 |
| ANIMALS | 19,47% | 1480 | 20,86% | 1709 | 21,08% | 1729 | 20,73% | 1837 | 18,83% | 1853 | 17,45% | 1739 | 16,57% | 1921 |
| MISUSE | 0,13% | 10 | 0,15% | 12 | 0,28% | 23 | 0,33% | 29 | 0,52% | 51 | 0,70% | 70 | 0,55% | 64 |
| HUMAN EMBRYOS/FOETUS | 0,55% | 42 | 0,81% | 66 | 0,70% | 57 | 0,78% | 69 | 0,72% | 71 | 0,60% | 60 | 0,57% | 66 |
| CIVIL APPLICATIONS | 0,00% | 0 | 0,00% | 0 | 0,12% | 10 | 0,14% | 12 | 0,18% | 18 | 0,12% | 12 | 0,09% | 10 |

The denominator is the number of proposals with ethical issues declared. The data present the sum of all categories, as the applicants could have declared none, one or more ethical issues.

visible, until the end of the framework period. Ethics issues in the categories 'Humans' and 'Human cells/tissues' are the next two most represented ethics issues claimed in the ethics self-assessment table. With some minor exceptions, their percentages remained stable during the entire analyzed period. Conversely, the frequency of ethics issues related to 'Animals' decreased over the framework period. Results indicate a very low frequency (below 1.5%) of the remaining ethics issues, including 'Dual use', 'Potential misuse', and 'Human embryos and foetuses', which had the lowest frequencies among ethics categories. The applicants used the category 'Exclusive focus on civil applications' the least (below 0.2%) (Table 1).

The percentage of the six most commonly used ethics issues categories ('Protection of personal data', 'Non-EU countries', 'Environment/health and safety issues', 'Humans', 'Human cells/tissues', 'Animals') reported by the applicants in self-assessment table of all submitted proposals, divided per MSCA actions over the years is shown in Fig 3. When self-declared ethics issues were divided per MSCA action, results showed similar trends of increasing frequency of self-declared ethics issues related to data protection and privacy, with the exception of the RISE action, where decreased frequency of this category was observed in years 2019–2020.

Comparable to the results shown in Table 1, the frequency of self-reported ethics issues related to non-EU countries substantially decreased after 2015 among all MSCA actions. The percentages of declared 'Environment, health and safety' ethics issues remained stable in all MSCA actions, except for IF, where an increasing trend was observed. Furthermore, the frequency of ethics issues category 'Humans' slightly increased in all MSCA actions over the analyzed years. However, results for the ethics category 'Human cells/tissues' remained stable for ITN, COFUND, and IF, while in the RISE action, a small increase was visible after 2016, similarly to the ethics category 'Animals'. In the IF action, there was the strongest decrease for the frequency of usage of category 'Animals' after 2018 (Fig 3).

Source data of frequencies and percentages for all ethics categories are presented in S1 File.

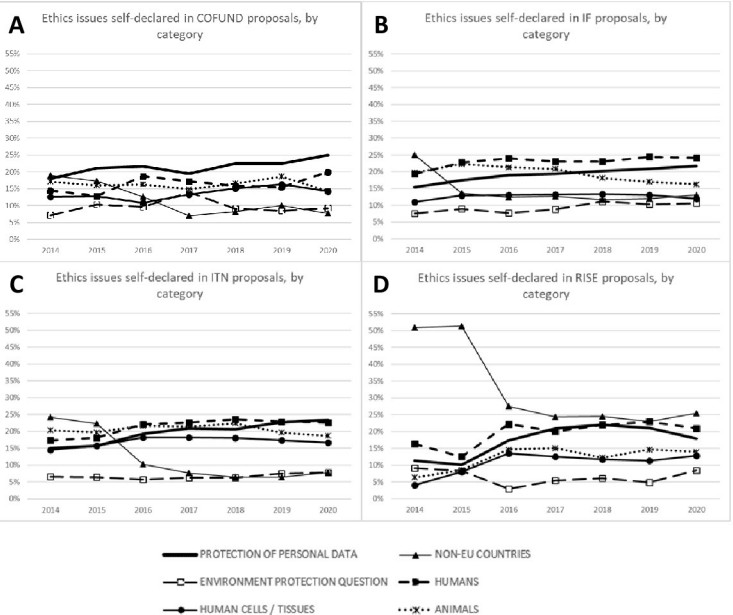

**Fig 3. Frequency of ethical issues reported by the applicants in self-assessment tables, divided per Marie Skłodowska-Curie actions (MSCA) submitted during the period 2014–2020.** The panels present proposals from MSCA COFUND (A), MSCA IF(B), MSCA ITN (C) and MSCA RISE (D). Diagrams show six most frequent ethics categories: Data; Non-EU, Humans, Human cells tissues, and Animals.

## Results of the ethics screening process by experts of the proposals on the main list

Ethics issues flagged by the experts in the ethics screening process are shown in Table 2. The table indicates that the ethics category 'Protection of personal data' is most represented during the entire H2020 period (2014–2020) and it almost equals the percentages represented for the ethics category 'Humans'. On the other side, ethics categories 'Exclusive focus on civil applications', 'Potential misuse', 'Dual use' and 'Human embryos/foetuses' are represented in values

**Table 2. Ethics issues flagged by the experts during ethics screening process.**

| Ethics issues declared in proposal, by category | 2014 | | 2015 | | 2016 | | 2017 | | 2018 | | 2019 | | 2020 | |
|---|---|---|---|---|---|---|---|---|---|---|---|---|---|---|
| | % | # | % | # | % | # | % | # | % | # | % | # | % | # |
| PROTECTION OF PERSONAL DATA | 23,26% | 1239 | 20,10% | 753 | 23,56% | 675 | 22,90% | 691 | 27,88% | 928 | 26,50% | 978 | 29,53% | 1484 |
| NON-EU COUNTRIES | 14,34% | 764 | 14,87% | 557 | 15,22% | 436 | 14,95% | 451 | 12,74% | 424 | 14,12% | 521 | 11,36% | 571 |
| ENVIRONMENT PROTECTION QUESTION | 6,83% | 364 | 10,41% | 390 | 13,72% | 393 | 17,50% | 528 | 15,17% | 505 | 17,37% | 641 | 19,42% | 976 |
| OTHER ETHICS ISSUES | 5,76% | 307 | 3,28% | 123 | 0,77% | 22 | 1,06% | 32 | 1,20% | 40 | 0,70% | 26 | 0,44% | 22 |
| DUAL USE | 0,75% | 40 | 1,33% | 50 | 0,87% | 25 | 0,36% | 11 | 0,18% | 6 | 0,33% | 12 | 0,38% | 19 |
| HUMANS | 22,83% | 1216 | 21,54% | 807 | 22,16% | 635 | 19,42% | 586 | 19,77% | 658 | 20,40% | 753 | 20,84% | 1047 |
| HUMAN CELLS / TISSUES | 7,81% | 416 | 8,38% | 314 | 8,80% | 252 | 7,95% | 240 | 8,80% | 293 | 7,04% | 260 | 6,29% | 316 |
| ANIMALS | 16,35% | 871 | 12,76% | 478 | 9,53% | 273 | 9,94% | 300 | 7,72% | 257 | 7,07% | 261 | 7,26% | 365 |
| MISUSE | 0,60% | 32 | 0,59% | 22 | 0,98% | 28 | 1,23% | 37 | 0,78% | 26 | 0,98% | 36 | 0,28% | 14 |
| HUMAN EMBRYOS/FOETUS | 1,46% | 78 | 0,67% | 25 | 0,38% | 11 | 0,36% | 11 | 0,45% | 15 | 0,27% | 10 | 0,10% | 5 |
| CIVIL APPLICATIONS | 0,00% | 0 | 0,00% | 0 | 0,00% | 0 | 0,00% | 0 | 0,00% | 0 | 0,00% | 0 | 0,00% | 0 |
| GENERAL | 0,00% | 0 | 6,06% | 227 | 4,01% | 115 | 4,31% | 130 | 5,29% | 176 | 5,23% | 193 | 4,10% | 206 |

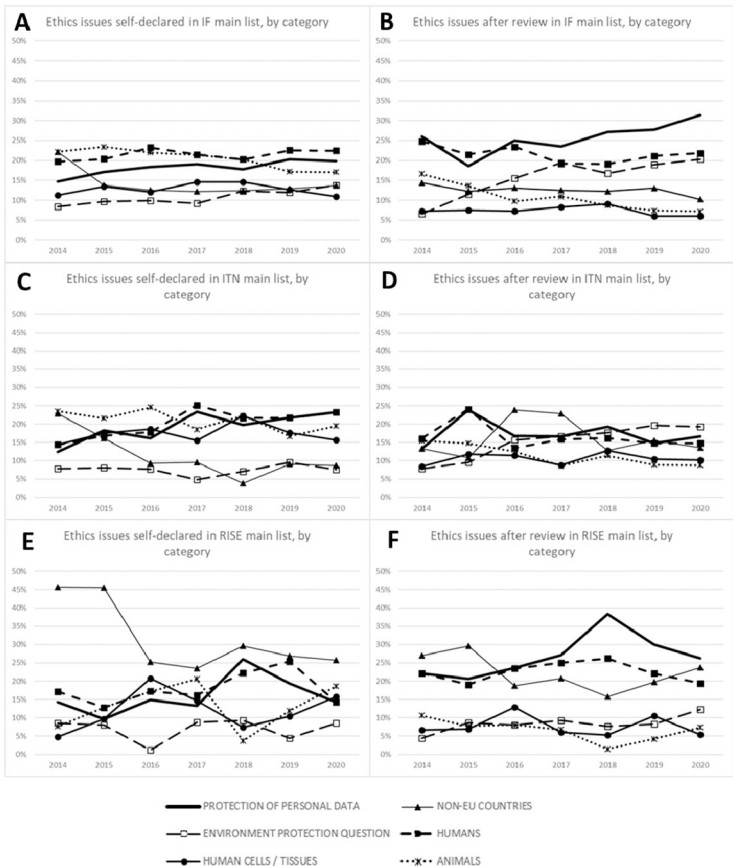

**Fig 4. Comparison of ethics issues declared by applicants during the ethics self-assessment with ethics issues declared from ethics experts conducting the ethics review of proposals on the MAIN lists.** Panels present percentages of six ethics categories (Data; Non-EU, Humans, Human cells tissues and Animals) of proposals on the MAIN lists and divided per MSCA actions. Ethics issues were identified either from applicants during the ethics self-assessment or from ethics experts during the ethics review procedure in the period of 2014–2020. Applicant's ethics self-assessment: A) MSCA IF, C) MSCA ITN and D) MSCA RISE. Ethics expert's ethics reviews: B) MSCA IF, D) MSCA ITN and F) MSCA RISE.

ranging from 0% for civil applications to 1,46% for 'Human embryos/foetuses' in 2014. Source data of frequencies and percentages for both applicants' self-assessment and ethics expert ethics reviews for all ethics categories are presented in S2 File.

The comparison between applicants' awareness of ethics issues as declared in the ethics-self assessment table with ethics issues flagged by experts during the ethics review is shown in Fig 4. Results are shown for the ITN, IF, and RISE MSCA actions. The graphical comparison of the six most frequently declared ethics issues categories is shown in Fig 4, while results of all ethics categories are provided in S2 File.

Results demonstrated a difference for IF and RISE MSCA actions for the category 'Protection of personal data', showing that applicants recognized significantly lower ethics issues in this category when compared to those detected from the side of ethics experts during the ethics review, similarly as for the category 'Environment, health and safety' in IF and ITN actions. Conversely, applicants tended to indicate more ethics issues in the self-assessment table in the category 'Animals', compared to the ethics experts' opinion during the ethics review.

**Table 3. Categories of "trajectory" describing the faith of all proposals applied under the MSCA H2020 during 2014–2020 in regard to ethics.**

| Trajectory of all proposals | Total |
|---|---|
| Inadmissible/ineligible | 600 |
| No self-declared ethics issues, no screening because of low ranking | 36323 |
| Self-declared ethics issues, no screening because of low ranking | 27419 |
| Proposals screened | 14725 |
| Total | 79067 |

## Complete ethics trajectory of all proposals submitted under all MSCA actions during the entire H2020 framework (2014–2020)

The outcome ('trajectory') of the complete ethics analysis of all proposals is shown in Table 3, while the outcome of screened proposals is shown in Table 4. From all submitted proposals, more than 50% did not declare any ethics issues in the ethics self-assessment table. Only a minor part of all proposals (around 19%) went to ethics screening, which is conducted only on the proposals which appear on the main and reserve lists. The ratio of self-declared ethics issue does not have any impact on the proposals scientific evaluation process, thus not influencing whether some proposal will appear on the main list or not.

From those that were retained on the main lists and were subject to an ethics review, the majority of the proposals passed the review process either with a conditional or unconditional ethics clearance. Only 458 were flagged for the ethics checks (Table 4). To our knowledge, there was no need to implement Ethics Audit during the entire H2020 framework period in the MSCA actions.

A detailed explanation of each outcome category mentioned in Tables 3 and 4 is shown in S3 File.

**Table 4. Categories of "trajectory" describing the final outcome of ethics screened proposals, showing what applicants declared, what happened during the screening/review and whether or not there was an ethics check proposed.**

| Trajectory of screened proposals | N | % |
|---|---|---|
| No self-declared ethics issues, incomplete data | 1565 | 10,63 |
| No self-declared ethics issues, screening resulted in "ethics clearance" | 3519 | 23,90 |
| No self-declared ethics issues, screening resulted in "conditional clearance" with requirements | 2637 | 17,91 |
| No self-declared ethics issues, screening resulted in "conditional clearance" with requirements, ethics check added | 84 | 0,57 |
| Self-declared ethics issues, incomplete data | 794 | 5,39 |
| Self-declared ethics issues, screening resulted in "ethics clearance" | 570 | 3,87 |
| Self-declared ethics issues, screening resulted in "conditional clearance" with requirements | 5180 | 35,18 |
| Self-declared ethics issues, incomplete data, screening resulted in "conditional clearance" with requirements, ethics check added | 1 | 0,01 |
| Self-declared ethics issues, screening resulted in "conditional clearance" with requirements, ethics check added | 373 | 2,53 |
| Self-declared ethics issues, screening resulted in "no clearance" | 1 | 0,01 |
| Total | 14724 | 100 |

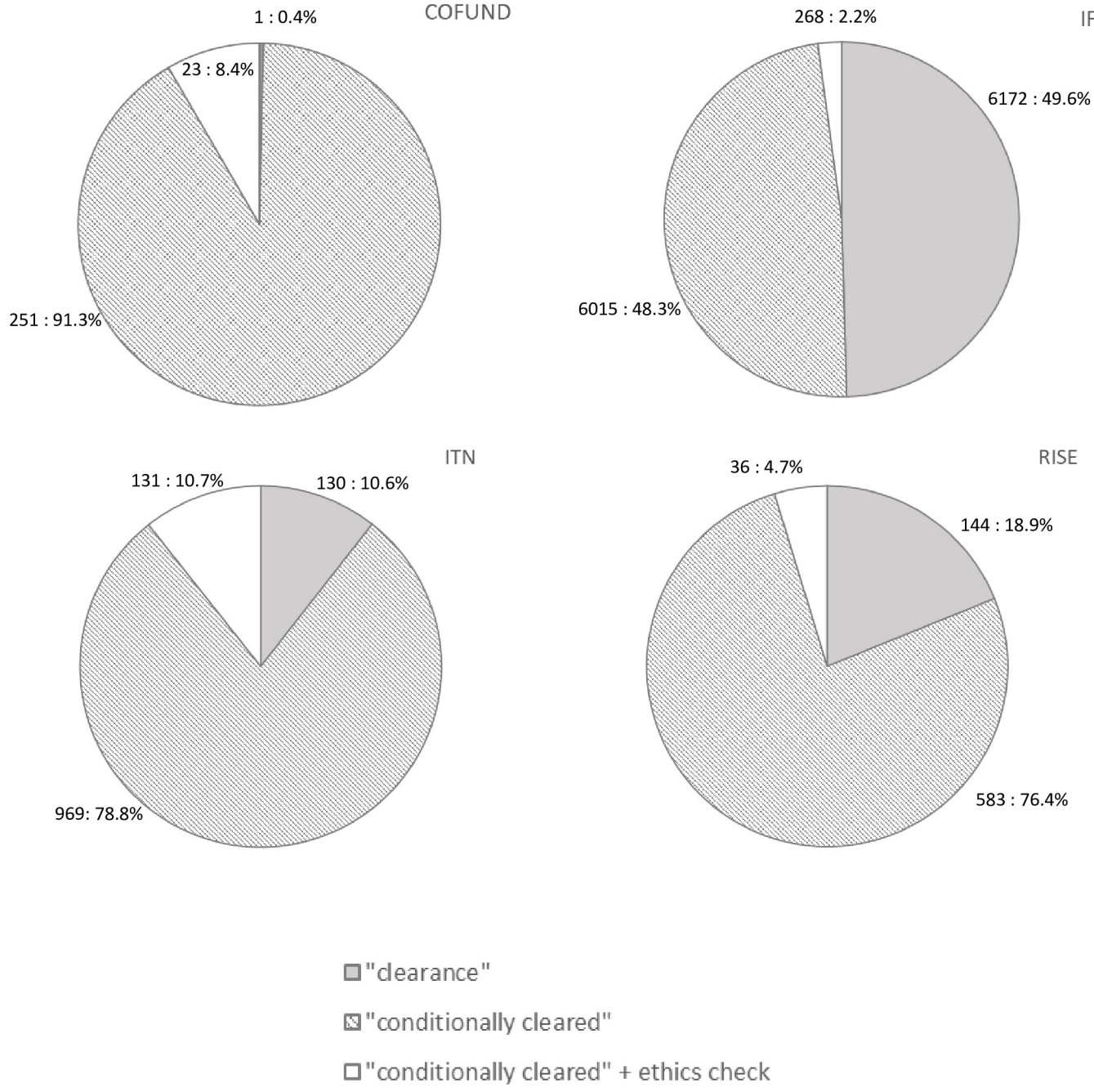

**Fig 5. Results of ethics screening for proposals on main lists for different MSCA actions: COFUND, IF, ITN and RISE.**

### Ethics "outcome" per action among proposals on the main list 2014–2020

Ethics screening of proposals on the main list (Fig 5) showed a difference in ethics outcome between IF as an individual research fellow application, compared to other MSCA actions where beneficiaries are institutions (consortium partners). Namely, an almost equal number of proposals received ethical clearance and conditional ethical clearance, while in other actions, this number was substantially lower, ranging from 0.4% for COFUND to 18.9% for

**Table 5. The number of proposals on main list that were flagged as needing an ethics check during project implementation.**

| Year | IF | | | ITN | | | COFUND | | | RISE | | |
|---|---|---|---|---|---|---|---|---|---|---|---|---|
| | no check | check | total | no check | check | total | no check | check | total | no check | check | total |
| 2014 | 1284 98,4% | 21 1,6% | 1305 100% | 111 91,7% | 10 8,3% | 121 100% | 2 8,7% | 21 91,3% | 23 100% | 83 98,8% | 1 1,2% | 84 100% |
| 2015 | 1124 96,6% | 39 3,4% | 1163 100% | 83 79,8% | 21 20,2% | 104 100% | 29 100,0% | 0 0,0% | 29 100% | 71 79,8% | 18 20,2% | 89 100% |
| 2016 | 1148 96,6% | 41 3,4% | 1189 100% | 99 90,8% | 10 9,2% | 109 100% | 33 100,0% | 0 0,0% | 33 100% | 81 95,3% | 4 4,7% | 85 100% |
| 2017 | 1293 95,8% | 56 4,2% | 1349 100% | 117 92,1% | 10 7,9% | 127 100% | 21 100,0% | 0 0,0% | 21 100% | 79 98,8% | 1 1,3% | 80 100% |
| 2018 | 1311 97,0% | 40 3,0% | 1351 100% | 102 84,3% | 19 15,7% | 121 100% | 28 100,0% | 0 0,0% | 28 100% | 69 93,2% | 5 6,8% | 74 100% |
| 2019 | 1458 98,8% | 18 1,2% | 1476 100% | 114 89,1% | 14 10,9% | 128 100% | 27 100,0% | 0 0,0% | 27 100% | 67 100,0% | 0 0,0% | 67 100% |
| 2020 | 1624 99,6% | 6 0,4% | 1630 100% | 139 93,9% | 9 6,1% | 148 100% | 43 100,0% | 0 0,0% | 43 100% | 73 98,6% | 1 1,4% | 74 100% |

the RISE. In opposite, the lowest level of proposals requiring ethics checks was represented among IF proposals.

## Ethics checks

The frequency of proposals on the main list that were flagged as needing an ethics check during the implementation phase, divided per MSCA actions during 2014–2020, is shown in the Table 5. Results are presented both as frequency of checks as well as the percentages from all proposals on the main lists.

Overall, among proposals with ethics screening, few proposals were sent for ethics check in all MSCA actions, except for COFUND at 2014 what led to the change of ethics review procedure after 2015.

## Discussion

Our findings indicate that MSCA actions are highly competitive funding lines, as only 14.3% of all applied proposals ended on the main funding lists during the entire H2020 period. The applicants for almost half of all submitted proposals and proposals retained on the main lists self-reported at least one type of ethics issue. Thus, the ethics appraisal process on those proposals is essential to adequately address key ethics questions for all projects granted funding [20].

Our results also show that the most common self-reported ethics issues categories among all submitted proposals were; 'Animals'; 'Environment/health & safety' 'Human cells / tissues'; 'Human beings' 'Non-EU countries' and 'Protection of personal data'. Our analysis did not permit going deeper into proposals with self-reported issues. Likely, there are differences in the perception of what constitutes an ethics issue between different researchers and institutions. It has been well documented that there are discrepancies between judgments of different ethics committees across Europe [21].

Ethics issues associated with the 'Protection of personal data' were among the most commonly used ethics issues, both by the applicants (based on ethics-self assessment tables) and by ethics experts (based on ethics experts' assessments). A higher frequency of that category was found in proposals submitted to IF and RISE actions than other actions. The increased frequency of its use in 2018 is likely linked to the implementation of the General Data Protection Regulation (GDPR) in the EU in May 2018. That effect was particularly notable among the proposals submitted to the RISE action, in which more than 35% of proposals reported data protection issues in 2018.

Data Protection Directive 95/46/EC [22] was in place in previous years, but the applicants and experts likely recognized it only as a directive and not a legal requirement in member

states. The introduction of GDPR [23] in 2018 during the Horizon 2020 Framework Programme brought in considerable heightened concern about data protection issues as member states sought to create legislation following the regulation. Ethics review panels recognized the need to ensure there were constituents who had the relevant expertise to apply the regulation, and so membership included more data protection legal experts and even, often newly appointed, Data Protection Officers (DPOs) [24].

It has already been suggested [20] that one of the main areas for possible "overkill" in ethics requirements in Horizon 2020 is data protection. That is why the screening of data protection issues, based on the GDPR requirements, should be separated from the ethics review. Namely, data protection issues are regulated by the laws of the EU law, Members States and/or the legal commitments of participating third countries.

Further, based on the results coming from the analyzed main lists, percentages of indicated ethics issues related to the category 'Humans' were frequent and equally detected both by applicants and ethics experts. Since previous publication showed no evidence that multiple ethics reviews enhance protections for human subjects [25] and since ethical concerns arising from this category are well recognized [26], current ethics assessment procedures in H2020 related to this category seems sufficient to appropriately deal with potential ethics issues which may arise from this category.

When we compared the frequency of issues related to ethics categories "Animals" and "Human cells/tissues" between applicants and ethics experts, we found that these categories were less often indicated during ethics reviews compared to the applicants' ethics-self assessments. Even though some challenges in using animals for research purposes still remain [27], it is less likely that applicants' awareness related to this category is more developed than among ethics experts, especially as nowadays research on animals is a subject of strictly arranged procedures [28,29]. The reason for this difference might be that in the phase of ethics self-assessment, applicants tend to over-use this category even if their research does not directly use animals or human cells/tissues. That is usually the case for proposals based on previously published data using animals or proposals that are a part of a larger institutional project, where applicants are not using new animals, but using tissues of animals for which all ethical issues have already been cleared.

Thus, slight differences between ethics issues recognized by applicants and ethics experts could be due to insufficient knowledge of the applicants regarding research ethics. This could be addressed by educational interventions. Graduate schools should include mandatory research ethics curricula, which should impact the knowledge of early-stage researchers. This education does not need to be delivered face-to-face. It has been shown that ethics education can be successfully delivered in innovative online formats [30]. Online education in ethics for applicants to European projects could also be created, in addition to the current guidance on how to complete ethics self-assessment [31].

Finally, our trajectory analysis of all ethics screened proposals appearing on main lists showed that only 3.1% of all screened proposals required additional ethics check. When divided per individual MSCA action, the analysis showed that most proposals received ethics clearance with or without additional requirements after that ethics review. The need for an additional ethics check ranged from 2.2% for IF to 10.7% for ITN. Such a low need for ethics checks may indicate that ethics assessment procedures implemented at the early phase of MSCA H2020 proposal applications, together with appropriate communication of project officers with applicants, reduce the need for ethics checks in the project implementation phase.

It needs to be emphasized that not all MSCA actions are the same. For example, the COFUND scheme differs from the other MSCA schemes in that the research topics are unknown at the time of application. The IT systems in place for evaluation do not properly

reflect this possibility and both options—marking all or none for ethics check—have been trialed. When all, or near all, have been marked, it led to extra unnecessary work and therefore the idea was dropped. The proper monitoring of ethics issues in COFUND takes place through compulsory ethics deliverables and is actively monitored by the REA throughout the project lifetime.

One of the study limitations is that we have based our main analysis on the "main list", but some of the proposals listed on the main list will not be funded eventually. However, this percentage of proposals not being funded from the main list is negligible and should not influence the validity of results. Also, Ethics Review in H2020 includes Ethics screening and Ethics Assessment (an in-depth analysis of the ethical issues of the proposals). A number of proposals which went on Ethics Assessment (different from Ethics Checks) are not possible to clearly detect in Corda and thus we did not present them in this work.

For ethics check analysis, the limitation is that our table does not show how many recommended ethics checks were conducted and what the outcome was of the ethics check procedure.

In conclusion, personal data protection is one of the most represented ethics categories indicated among MSCA actions, especially after 2018 when GDPR was introduced. This ethics category may exhaust ethics assessment efforts and may lead to "overkills" in ethics requirements. A potential solution to this problem may be excluding the majority of personal data protection assessment from the ethics assessment, except for parts which are directly related to ethics like "Informed consent procedures", in a separate process that should involve specialized experts in personal data protection. A gap in understanding of ethics issues between applicants and reviewers' points to the necessity to further educate researchers on research ethics issues.

## Supporting information

**S1 File. Frequencies and percentages of all ethics issues categories reported by the applicants in self-assessment table of all submitted proposals, divided per MSCA actions from the entire European HORIZON 2020 research and innovation program (2014–2020).** (DOCX)

**S2 File. The comparison between applicants' awareness of ethics issues as declared in the ethics-self assessment table ("self-declared") with ethics issues flagged by experts during the ethics review ("after review") on the main lists, divided per MSCA actions from the entire European HORIZON 2020 research and innovation program (2014–2020).** Data are shown as frequencies and percentages of the total number of proposals with declared ethics issues. (DOCX)

**S3 File. Detailed explanation of each outcome category as shown in Tables 3 and 4.** (DOCX)

## Acknowledgments

Authors would like to thank Research Executive Agency of the European Commission for providing access to pooled anonymized data from the agencies database "Corda".

**Disclaimer**: All views expressed in this article are strictly those of the article authors and may in no circumstances be regarded as an official position of the Research Executive Agency or the European Commission.

## Author Contributions

**Conceptualization:** Livia Puljak, Zvonimir Koporc.

**Data curation:** Ilse De Waele, Livia Puljak, Zvonimir Koporc.

**Formal analysis:** Ilse De Waele, Zvonimir Koporc.

**Funding acquisition:** Zvonimir Koporc.

**Investigation:** Ilse De Waele, David Wizel, Livia Puljak, Zvonimir Koporc.

**Methodology:** Ilse De Waele, Livia Puljak, Zvonimir Koporc.

**Project administration:** Zvonimir Koporc.

**Resources:** Zvonimir Koporc.

**Software:** Ilse De Waele, Zvonimir Koporc.

**Supervision:** David Wizel, Livia Puljak, Zvonimir Koporc.

**Validation:** Ilse De Waele, David Wizel, Livia Puljak, Zvonimir Koporc.

**Visualization:** Ilse De Waele, Zvonimir Koporc.

**Writing – original draft:** Livia Puljak, Zvonimir Koporc.

**Writing – review & editing:** Ilse De Waele, David Wizel, Livia Puljak, Zvonimir Koporc.

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
