## [Decision Letter · Decision Letter 0]

6 Sep 2021

PONE-D-21-15555Ethics appraisal procedure in 79,670 Marie Skłodowska-Curie proposals from the entire European HORIZON 2020 research and innovation program (2014 - 2020): a retrospective analysisPLOS ONE

Dear Dr. Koporc,

Thank you for submitting your manuscript to PLOS ONE. After careful consideration, we feel that it has merit but does not fully meet PLOS ONE’s publication criteria as it currently stands. Therefore, we invite you to submit a revised version of the manuscript that addresses the points raised during the review process. There are just a few comments that you must address in order to reconsider my decision on this manuscript. However, please be very careful to address all them, and to provide suitable and complete responses and rationales to the Reviewer's comments.

We look forward to receiving your revised manuscript.

Kind regards,

Sergio A. Useche, Ph.D.

Academic Editor

PLOS ONE

2. Please do not include funding sources in the abstract or anywhere else in the manuscript file. Funding information should only be entered in the financial disclosure section of the submission system. https://journals.plos.org/plosone/s/submission-guidelines#loc-acknowledgments.

4. Thank you for stating the following in the Acknowledgments/Funding Section of your manuscript:

“This study was funded by the project Promoting integrity in the use of research results in evidence-based policy: a focus on non – medical research (PRO-RES) funded by the EU (H2020-SwafS-2016-17). We are grateful to the Research Executive Agency for providing anonymous data about the outcomes of the ethics evaluation of the submitted research proposals.”

“This study was funded by the project Promoting integrity in the use of research results in evidence-based policy: a focus on non – medical research (PRO-RES) funded by the EU (H2020-SwafS-2016-17). Authors are grateful to the Research Executive Agency for providing anonymous data about the outcomes of the ethics evaluation of the submitted research proposals.”

6. We note that you have indicated that data from this study are available upon request. PLOS only allows data to be available upon request if there are legal or ethical restrictions on sharing data publicly. For more information on unacceptable data access restrictions, please see http://journals.plos.org/plosone/s/data-availability#loc-unacceptable-data-access-restrictions.

7. Please provide additional details regarding participant consent. In the ethics statement in the Methods and online submission information, please clarify whether the authors of the proposals accessed for your study agreed to have their proposals used in research. Even if the proposals were anonymized, it may be easy to identify their authors' identity.

If a data protection committee or research ethics board assessed your study and waived a requirement for informed consent, please state so in your ethics statement.

Additional Editor Comments (if provided):

Reviewers' comments:

Reviewer's Responses to Questions

**Comments to the Author**

1. Is the manuscript technically sound, and do the data support the conclusions?

Reviewer #1: Yes

2. Has the statistical analysis been performed appropriately and rigorously? 

Reviewer #1: Yes

3. Have the authors made all data underlying the findings in their manuscript fully available?

Reviewer #1: Yes

4. Is the manuscript presented in an intelligible fashion and written in standard English?

Reviewer #1: Yes

5. Review Comments to the Author

Reviewer #1: I recommend this paper for publication. It offers a comprehensive review of an ethics process to determine its efficacy , highlighting gaps between applicant and expert understandings and offers suitable recommendations. The conclusions appear valid and the recommendations, limitations and further work are well highlighted.

A few very minor comments:

It would help to cross reference where key definitions are placed to enable easier reading of the main body of the paper eg. paragraph 1 of the results section refers to the "main list" and it was not initially clear how this list was derived.

Also in the results section there is a disparity in the figures (probably a typographical error). Paragraph 1 states 79670 proposals and paragraph 2 states 79067.

Finally in the discussion the statement "undoubtedly linked to the implementation of the General Data Protection Regulation" would benefit from more caveated language as the use of "undoubtedly" without supporting empirical evidence (eg. data linked to increased number of data protection experts on the list of investigators) is a little strong.

6. PLOS authors have the option to publish the peer review history of their article (what does this mean?). If published, this will include your full peer review and any attached files.

Reviewer #1: **Yes: **Dr Allison Gardner

---

## [Author Response · Author response to Decision Letter 0]

14 Sep 2021

-Response to Reviewers - provided as a separate document in the section Files, for the 1st revision

-Response to the 2nd revision is provided as an additional separate document (submitted on 14 September 2021)

---

## [Decision Letter · Decision Letter 1]

22 Oct 2021

Ethics appraisal procedure in 79,670 Marie Skłodowska-Curie proposals from the entire European HORIZON 2020 research and innovation program (2014 - 2020): a retrospective analysis

PONE-D-21-15555R1

Dear Dr. Koporc,

We’re pleased to inform you that your manuscript has been judged scientifically suitable for publication and will be formally accepted for publication once it meets all outstanding technical requirements.

Kind regards,

Sergio A. Useche, Ph.D.

Academic Editor

PLOS ONE

Additional Editor Comments (optional):

Reviewers' comments:

Reviewer's Responses to Questions

**Comments to the Author**

1. If the authors have adequately addressed your comments raised in a previous round of review and you feel that this manuscript is now acceptable for publication, you may indicate that here to bypass the “Comments to the Author” section, enter your conflict of interest statement in the “Confidential to Editor” section, and submit your "Accept" recommendation.

Reviewer #1: All comments have been addressed

2. Is the manuscript technically sound, and do the data support the conclusions?

Reviewer #1: Yes

3. Has the statistical analysis been performed appropriately and rigorously? 

Reviewer #1: Yes

4. Have the authors made all data underlying the findings in their manuscript fully available?

Reviewer #1: Yes

5. Is the manuscript presented in an intelligible fashion and written in standard English?

Reviewer #1: Yes

6. Review Comments to the Author

Reviewer #1: (No Response)

7. PLOS authors have the option to publish the peer review history of their article (what does this mean?). If published, this will include your full peer review and any attached files.

Reviewer #1: **Yes: **Dr Allison Gardner

---

## [Editor Report · Acceptance letter]

26 Oct 2021

PONE-D-21-15555R1 

Ethics appraisal procedure in 79,670 Marie Skłodowska-Curie proposals from the entire European HORIZON 2020 research and innovation program (2014 - 2020): a retrospective analysis 

Dear Dr. Koporc:

I'm pleased to inform you that your manuscript has been deemed suitable for publication in PLOS ONE. Congratulations! Your manuscript is now with our production department. 

Kind regards, 

on behalf of

Dr. Sergio A. Useche 

Academic Editor

PLOS ONE